# Three-Dimensional Printing and Fracture Mapping in Pelvic and Acetabular Fractures: A Systematic Review and Meta-Analysis

**DOI:** 10.3390/jcm11185258

**Published:** 2022-09-06

**Authors:** Alvin Kai-Xing Lee, Tsung-Li Lin, Chin-Jung Hsu, Yi-Chin Fong, Hsien-Te Chen, Chun-Hao Tsai

**Affiliations:** 1Department of Education, China Medical University Hospital, Taichung 404, Taiwan; 2Department of Orthopedics Surgery, China Medical University Hospital, Taichung 404, Taiwan; 3Department of Sports Medicine, China Medical University, Taichung 404, Taiwan; 4Graduate Institute of Biomedical Sciences, China Medical University, Taichung 404, Taiwan; 5School of Chinese Medicine, China Medical University, Taichung 404, Taiwan; 6Department of Orthopedics Surgery, China Medical University Hospital Beigang Branch, Yunlin 651, Taiwan; 7Spine Center, China Medical University Hospital, Taichung 404, Taiwan

**Keywords:** three-dimensional printing, fracture mapping, meta-analysis

## Abstract

Three-dimensional printing and fracture mapping technology is gaining popularity for preoperative planning of fractures. The aim of this meta-analysis is to further understand for the effects of 3D printing and fracture mapping on intraoperative parameters, postoperative complications, and functional recovery on pelvic and acetabular fractures. The PubMed, Embase, Cochrane and Web of Science databases were systematically searched for articles according to established criteria. A total of 17 studies were included in this study, of which 3 were RCTs, with a total of 889 patients, including 458 patients treated by traditional open reduction and internal fixation methods and 431 patients treated using 3D printing strategies. It was revealed that three-dimensional printing and fracture mapping reduced intraoperative surgical duration (RoM 0.74; 95% CI; 0.66–0.83; I^2^ = 93%), and blood loss (RoM 0.71; 95% CI; 0.63–0.81; I^2^ = 71%). as compared to traditional surgical approaches. In addition, there was significantly lower exposure to intraoperative imaging (RoM 0.36; 95% CI; 0.17–0.76; I^2^ = 99%), significantly lower postoperative complications (OR 0.42; 95% CI; 0.22–0.78; I^2^ = 9%) and significantly higher excellent/good reduction (OR 1.53; 95% CI; 1.08–2.17; I^2^ = 0%) in the three-dimensional printing and fracture mapping group. Further stratification results with only prospective studies showed similar trends. Three-dimensional printing and fracture mapping technology has potential in enhancing treatment of complex fractures by improving surgical related factors and functional outcomes and therefore could be considered as a viable tool for future clinical applications.

## 1. Introduction

The overall incidence of pelvic and acetabular fractures is increasing exponentially due to our aging population and increasing cases of road crashes [1,2]. The occurrence of acetabular fractures is bimodal and there are two age peaks, with each age group having its own unique causes of acetabular fractures [3]. It is a well-known fact that the younger population sustain from acetabular fractures due to high-velocity trauma (such as road crashes) whilst elderly patients tend to sustain from low-impact trauma (such as falls) [4]. Regardless of age, the treatment of pelvic and acetabular fractures remains one of the most challenging tasks for an orthopedic surgeon due to the complex structures of the pelvis and limited surgical approaches [5]. A good reduction requires the surgeon to fully understand the extent of the fractures and to choose a suitable surgical approach to obtain an anatomic reduction of the bone fragments [6]. To make things complicated, each case is unique and thus there is no single anatomically correct implant to fix all cases [7,8]. Currently, there are no standard guidelines on the best approach to pelvic and acetabular fractures and thus clinical reduction depends significantly on the experiences of the lead surgeon. More often than not, the patient is exposed to excessive intraoperative radiation exposure due to the need for plate contouring and finding the proper screw length for optimal reduction [9]. This would also lead to prolonged surgical duration and increased blood loss.

Preoperative planning is essential and crucial to ensure a good surgery outcome [10,11]. In recent years, advancements in the three-dimensional (3D) reconstruction of radiological images has further fine-tuned the visualization of surgical planning [12]. With the development of modern technology such as rapid prototyping or otherwise known as 3D printing, we are now able to generate life-size 3D models from computed tomography (CT) images of patients [13]. This 3D model provides surgeons with a direct and interactive display of the fractures, and thus allows surgeons to prepare individualized treatment strategies for patients [14,15]. Since then, 3D models have also been used in training junior surgeons and medical students either in understanding fracture patterns or for anatomy education. In clinical settings, these models were often used for inter- and intra-team communications and rehearsals [16,17]. There were also multiple reports showing increased patient satisfaction and understanding when the 3D models were used for physician–patient communication [18]. In 2018, Lal et al. published a literature review showing that 3D printing is gaining popularity in orthopedics surgery, with 3D printing being used on acromion, clavicle, humerus, radius, hand, acetabulum, pelvic, and femur fractures [8]. Due to the potential impact that 3D printing can have on patient safety and functional outcome, we are seeing increasing reports of individualized treatment strategies for complex fractures. Since then, 3D printing approaches have evolved and improved to include virtual fracture reduction simulation using non-mirroring techniques and algorithms to determine fracture mapping and classification [19,20]. Therefore, an updated meta-analysis will be helpful to validate its feasibility in both pelvic and acetabular fractures, which are considered complex fractures in orthopedic trauma. In-depth discussions regarding the different techniques are provided in the discussion below.

In this meta-analysis and systematic review, we systematically conducted a quantitative review of 3D printing in pelvic and acetabular trauma. Therefore, the aim of this meta-analysis is to further understand the effects of 3D printing and fracture mapping on intraoperative parameters, postoperative complications, and functional recovery on pelvic and acetabular fractures.

## 2. Materials and Methods

This systematic review and meta-analysis were conducted according to the Cochrane Guidelines and are reported in agreement with the Preferred Reporting Items for Systemic Reviews and Meta-Analysis (PRISMA) guidelines. Ethical approval was not required for this study.

### 2.1. Search Strategy

This systematic review and meta-analysis were performed on Pubmed, EMBASE, and Web of Science databases. The set of search terms used were manufacturing processes (3D printing; rapid prototyping; additive manufacturing; fracture mapping), acetabular fracture, or pelvic fracture. Data were collected using three modalities: electronic database tracking based on keywords, tracking based on the references of full-text screened studies, and tracking based on citations of full-text screened studies. The time frame for the literature search was January 2010 to Jan 2022 due to the availability of 3D printing and mapping technology. Studies in English and Mandarin were considered.

### 2.2. Study Selection

Two authors (Lee and Tsai) independently conducted title, abstract and full-text screenings to determine article eligibility. The inclusion criteria were as follows: (1) pelvic/acetabular fracture; (2) 3D printing, pre-contouring of plates, or 3D simulation; and (3) clinical trials, reports, and series with controls. The exclusion criteria were as follows: (1) animal studies; (2) abstract-only papers as preceding papers, conference, editorial, and author response theses and books; (3) articles without available full text; and (4) clinical trials, reports, and series without controls. Duplicated studies were removed using an excel spreadsheet and Endnote. A third reviewer (Chen) was consulted in the case of any discrepancy. No additional papers on the topic were found within the references of the screened articles.

### 2.3. Study Identification

The systemic search initially retrieved 254 potentially relevant studies with 73, 134, and 47 studies from Pubmed, EMBASE, and Web of Science, respectively. A total of 92 studies were removed after a duplication check and 115 studies were excluded after a review of the title and abstract. After which, 47 studies were assessed for full-text eligibility and of which, 25 studies were found to have wrong study designs (non-comparative studies without control groups) and 5 studies were found to have wrong interventions (studies with non-surgical interventions). In total, 17 full-text articles, of which, 3 were randomized controlled trials (RCTs), 3 were prospective studies and 11 were retrospective studies. A flow chart of the study selection process was shown in Figure 1. 

### 2.4. Data Extraction

Two authors (Hsu and Lin) independently extracted the data from each accepted study according to a pre-established protocol and a third author (Fong) verified the data extraction sheet. In this study, we extracted characteristics of each study (authors, publication, year country, and study design), characteristics of patients (age, gender, duration of follow-up), and key outcomes (operation duration (minutes), blood loss (mL), times of intraoperative image, number of complications and excellent/good reduction).

### 2.5. Study Characteristics

The characteristics of the 17 studies were as presented in Table 1. 11 out of 17 of the studies were reported to be from China, with 3 from Taiwan, 2 from India, and 1 from Ireland. All studies included 3D printing and fracture mapping with control groups. A total of 889 patients were included in the studies, including 458 patients treated by traditional open reduction and internal fixation methods and 431 patients treated using 3D mapping and printing strategies. The mean ages of the patients ranged from 33 to 52 and more than half of the patients were males. The mean follow-up period ranged from 1 month to 10 years as reported in one study. Three types of classifications were used in the studies, namely the Tile classifications, Young–Burgess classifications, and AO/OTA classifications for pelvic and acetabular fractures.

### 2.6. Risk of Bias Assessment

The risk of bias was assessed according to the Cochrane guidelines for randomized controlled trials and non-randomized controlled trials to be categorized as high, unclear, and low risk of bias. For the randomized controlled trials, the risk of bias was evaluated according to the following five domains: randomization process, deviations from intended interventions, missing outcome data, measurement of the outcome, and selection of the reported results. For the non-randomized trials including the prospective and retrospective studies, the risk of bias was evaluated according to the following seven domains: confounding factors, selection of participants, classification of interventions, deviation from intended interventions, missing outcome data, measurement of the outcome, and selection of the reported results. A score was given for each domain, and the addition of the scores resulted in a total score, with higher scores indicating higher quality of the articles.

Figure 2 provides a risk of bias summary for each selected study. There were concerns raised from the randomization processes of the 3 RCTs leading to unclear risks of randomization. However, the 3 RCTs scored well in the area aspects of the risk of bias assessment, thus they were rated as having a low risk of bias overall. For the non-RCTs, the bias and limitations of each study were clearly listed out and eliminated, thus they were rated as having a low risk of bias.

### 2.7. Data Synthesis and Statistical Analysis

Continuous outcomes such as operation duration (minutes), blood loss (mL), and times of intraoperative imaging were presented as mean ± SD. According to the Cochrane Handbook of Systematic Reviews of Interventions, we pooled the mean and SD in the 3D printing group in the study by Hsu et al., which reported results separately for patients with anterior or posterior column involvement. In the study by Huang et al., which only reported median and interquartile, we used the median to approximate the mean and used the interquartile range to approximate the SD. In addition, the ratio of means (RoM) was used to combine qualitative results from the various studies which were determined by the average outcome measurement for the 3D printing groups as compared with the traditional intervention groups. The quality of the repair was a dichotomous variable with excellent/good, which was evaluated by doctors using Matta scorings. For the number of postoperative complications and quality of repair, we used odds ratios (ORs) with 95% confidence intervals (CI) to standardize the reported results across studies.

A forest plot was used to present the summary estimate with 95% CI with the random effects models. A two-tailed *p*-value less than 0.05 was considered significant. Statistical heterogeneity was assessed using the χ2 Cochran Q test and I^2^ statistics. We used funnel plots to assess publication bias for the outcomes including more than 10 studies and used Egger’s test to detect bias statistically. All statistical analyses were performed by using R Core Team 2021 software.

### 2.8. Publication Bias

Funnel plots combined with Egger’s tests were used to assess for potential publication bias (Figure 3). Visual inspection of the funnel plot revealed some asymmetry among studies measuring operation duration, blood loss, number of complications, and excellent/good reduction. However, Egger’s test results did not show statistical differences. Therefore, the impact of potential publication bias on effect size was deemed less likely but cannot be totally ruled out.

## 3. Results

The effects of 3D printing and fracture mapping technology used for the treatment of pelvic and acetabular fractures are as shown in Figure 4. In addition, we further stratified the RCTs and prospective studies and evaluated for the effects of 3D printing and fracture mapping technology on the various intraoperative parameters, quality of reduction, and functional recovery if applicable (Figure 5).

The heterogeneity was listed in the respective figures.

### 3.1. Operation Duration

All 17 studies reported on the duration of surgeries. There were two distinct types of reported statistics for the duration of surgeries. Several articles reported on the total duration whilst some reported on the duration per screw. Therefore, we used the ratio of means (RoM) to evaluate the percentage of time saved per surgery. Our meta-analysis showed that the 3D printing group had 26% shorter surgical duration as compared to the traditional group (Figure 4A; RoM 0.74; 95% CI; 0.66–0.83; I^2^ = 93%). In addition, six RCTs and prospective studies reported on surgical duration and results showed that the 3D printing group had a 23% shorter surgical duration as compared to the traditional group (Figure 5A; RoM 0.77; 95% CI; 0.58–1.03; I^2^ = 95%).

### 3.2. Blood Loss

Twelve studies reported blood loss during operation. Our meta-analysis showed that the 3D printing group had 29% lower blood loss as compared to the traditional group (Figure 4B; RoM 0.71; 95% CI; 0.63–0.81; I^2^ = 71%). In addition, five RCTs and prospective studies reported on blood loss and results showed that the 3D printing group had 21% lower blood loss as compared to the traditional group (Figure 5B; RoM 0.79; 95% CI; 0.59–1.06; I^2^ = 81%).

### 3.3. Intraoperative X-ray Exposure

Six studies reported on the amount of intraoperative X-ray exposure. Our meta-analysis showed that the 3D printing group had 74% lesser X-ray exposures as compared to the traditional group (Figure 4C; RoM 0.36; 95% CI; 0.17–0.76; I^2^ = 99%). In addition, the results from three RCTs and prospective studies showed that the 3D printing group had significantly lesser X-ray exposure as compared to the traditional group (Figure 5C; RoM 0.32; 95% CI; 0.11–0.93; I^2^ = 97%).

### 3.4. Number of Postoperative Complications

Twelve studies reported on the number of postoperative complications. Our meta-analysis showed that the 3D printing group had significantly reduced postoperative complications as compared to the traditional group (Figure 4D; OR 0.42; 95% CI; 0.22–0.78; I^2^ = 9%). All of the articles reported that 3D printing led to reduced postoperative complications. Similarly, the RCTs and prospective studies showed significantly reduced postoperative complications as compared to the traditional group (Figure 5D; OR 0.23; 95% CI; 0.10–0.55; I^2^ = 0%). From the RCTs and prospective studies, there was a total of 30 patients with complications in the traditional group as compared to 8 in the 3D printing group. In the 3D printing group, there were two heterotrophic ossifications, one soft tissue inflammation, one common fibular nerve injury, one unspecified iatrogenic neurological symptom, two traumatic arthritis, and one screw loosening. In the traditional group, there were three heterotrophic ossifications, nine inflammations, one heterotrophic ossification, six unspecified iatrogenic neurological symptoms, one sciatic nerve injury, two common fibular nerve injury, two obturator nerve injuries, five post-traumatic arthritis, and one deep vein thrombosis.

### 3.5. Excellent/Good Reduction

All 17 studies reported a number of excellent/good reductions. Our meta-analysis showed that the 3D printing group had significantly higher excellent/good reduction as compared to the traditional group (Figure 4E; OR 1.53; 95% CI; 1.08–2.17; I^2^ = 0%). An important factor to note in this evaluation was that we could not extract the entire information as a continuous variable and were only able to extract the proportion of patients with excellent/good ratings from all the studies. Similarly, the RCTs and prospective studies showed that the 3D printing group had significantly higher excellent/good reduction as compared to the traditional group (Figure 5E; OR 2.00; 95% CI; 1.07–3.73; I^2^ = 0%).

## 4. Discussion

The emergence and development of 3D printing and simulation technology are considered a significant paradigm shift for the orthopedic trauma community and are expected to bring about major changes to the treatment protocol of traumatic fractures, especially so for those with complex anatomies such as the pelvis or acetabular fractures [26,27]. This meta-analysis and systematic review were performed with an aim to establish whether 3D printing had a role to play in influencing (i) preoperative planning, (ii) intraoperative indices, and (iii) postoperative complications and quality of the repair. We arranged the selected articles in chronological order in Table 1 so as to obtain a clearer understanding of the development of 3D printing applications in pelvic and acetabular surgeries. As seen, the first 3D printing-related retrospective publication was in 2016, with increasing numbers of larger-scale publications noted as the years passed. This hinted at the potential implications that 3D printing can have on pelvic and acetabular fractures. In addition, we could see an interesting development in 3D printing technology. The first few studies from 2016 to 2018 use direct printing of the fractured pelvis model for preoperative preparation, followed by a surge in mirroring of the healthy contralateral pelvis technique in 2018. The latest trend in 3D printing was the incorporation of virtual fracture reduction simulation and virtual plate and screw placement simulation [22,23]. Furthermore, we could see novel ideas being introduced into the various studies such as printing of external template guides and even 3D printing of patient-specific plates and screws. However, there are clear guidelines for local regulations for patient-specific implants, which would not be further discussed in this paper. From the trends above, we could gradually see maturation in 3D printing techniques for pelvic and acetabular fractures, accompanied by evolving technology in virtual reality and simulation [28]. Currently, most of the studies were from the Asia region, including China and India.

Fractures of the pelvis and acetabulum are considered complex fractures due to the difficult-to-approach anatomy and composite anatomical structures [29,30]. Moreover, pelvis fractures are often caused by trauma and could potentially involve articular surfaces, thus there is a need for careful consideration and a clear understanding of the fracture pattern prior to making a treatment strategy [24]. Conventional imaging modalities such as X-ray and CT scanning are still unable to show osseous anatomy clearly; therefore, surgeons have to depend on their experiences and spatial imaginations to provide proper diagnosis and treatment strategies for the patients [25,31]. The main finding of this meta-analysis was that there were more pros from the 3D printing group in terms of surgical duration, reduced blood loss, number of X-ray exposures, postoperative complications, and improved reductions and functional recoveries. However, statistically wise, there were also individual publications amongst our meta-analysis stating otherwise. Chen et al. showed that the application of mirroring 3D printing for pelvic fractures led to a significant decrease in surgical duration and blood loss but no significant improvement in postoperative reduction (Matta scores) and functional recoveries (Merle d’Aubigné hip scores) [7]. In addition, Li et al. showed that amongst all the parameters, mirroring 3D printing for pelvic fractures only led to significant improvements in reducing intraoperative instrumentation timings [2]. The study by Downey et al. also confirmed that 3D printing of fractured pelvis models for preoperative planning had no significant impacts on objective factors such as surgical-related outcomes, reduction qualities, and functional recoveries (EQ-5D-5L scores) [10]. However, a subjective questionnaire was also completed to seek the surgeon’s experiences when using the 3D models. Results indicated that the 3D models contributed to improved patient consenting and improved understanding and physician–patient communication expectations. Furthermore, the surgeons felt that the 3D models contributed to improved inter- and intra-specialty communication and enhanced overall team confidence during the surgery itself. Even though the models had no significant impact on treatment strategies and intra-operative parameters, the authors further elaborated that the 3D models were able to improve intra- and inter-observer reliability in fracture classifications, especially if compared to CT imaging. It was reported in their study that without the aid of 3D models, experienced surgeons had an 11% accuracy in classifying acetabular fractures using X-ray and 30% accuracy using CT imaging. The 3D models were able to provide us with details that could not be easily noticed by using conventional CT imaging. As seen from our results above, all of the studies applied CT for all the patients, therefore the models could potentially provide us with a clearer understanding of the fracture patterns, thus influencing our treatment strategies.

A patient-specific and precise preoperative planning is crucial for pelvic and acetabular fractures. Currently, there are no standard guidelines and protocols as to which approach is the most ideal for certain types of pelvic or acetabular fractures [32]. Surgical approaches were often determined by the most senior surgeon by using traditional imaging modalities and his or her preferences and experiences [21]. As reported in all of the studies, a 3D model is able to enhance our understanding of complex fracture patterns, therefore allowing us to evaluate and decide on suitable treatment strategies for the restoration of the fracture. Taking it a step further, a recent publication by Yang et al. showed that fracture mapping could be achieved using 3D reconstructed models in order to allow us to better understand fracture fault lines which would then have a role to play in influencing surgeons’ decisions in surgical approaches and reduction and fixation strategies [33]. This clearly showed that we could leverage developing technologies to come up with novel ways to attempt to enhance the standard of care for patients.

Avascular necrosis, traumatic arthritis, and neural injury remained the top three postoperative complications for the 3D printing group. This is an interesting finding as the complications mainly involve soft tissues, of which these soft tissues were usually removed via thresholding during image processing. Therefore, even though we might be able to improve the quality of bone reduction via 3D printing, there is still a risk of soft tissue injury, thus leading to poor quality of life in the future. Indeed, the chance of postoperative complications in the traditional group was reported to be 8 to 10% as compared to only 4 to 5% for the 3D printing group. However, such injuries are concerning as subsequent revision surgeries might be required which are not further reported in all the studies. Li et al. was the only study to have included and considered soft tissues such as vessels and muscular structures in their 3D printing model [18]. Future 3D printing applications should attempt to incorporate the evaluation of soft tissues in their preoperative planning. The fracture mapping technique developed by Yang et al. could be a potential tool as the condition of soft tissues was considered in their planning [33].

Our meta-analysis has several limitations. Firstly, 11 out of the 17 articles were retrospective in nature, thus making our results more prone to both systematic and random errors due to uncontrolled bias. In this aspect, we could not control the bias that might have occurred due to differences in severity and classification of the injury as indicated in Table 1. Each publication has its own inclusion factors which involve different fracture patterns and severity. Furthermore, each clinical institution has its own standard operating procedure, such as external fixation or traction at the emergency department, which may cause differences in measured intraoperative outcomes. We are also unable to eliminate the level of surgical experiences between each institution. Therefore, other than using the ratio of means and odds ratio to attempt to mitigate the level of bias, future studies should include a larger number of cases in randomized controlled trials to further improve results reliability. Secondly, most of the studies have no long-term follow-ups regarding the functional recovery and reduction quality of patients. Thus, we proposed that there should be a generalized agreement regarding the parameters to be recorded and reported for future prospective studies. Classification-wise, future classifications should attempt to record the classifications before and after simulation to determine if simulation and 3D printing are able to enhance our understanding of fracture patterns. In addition, the classifications should include all available and common types of classifications such as Tile classifications, Judet–Letournel classification, and AO/OTA classifications for pelvic or acetabular fractures. We had listed the types of key operative outcomes in Table 1 that could be used as a platform for consideration during future studies. Future articles should also attempt to report on short-term complications such as localized soft tissue infection, and on long-term complications such as screw loosening and non-union so as to give readers a complete overview regarding the efficacy of 3D printing. By stating so, it is thus advised that the patients should have a minimum follow-up of at least 6 months to 2 years for observations of long-term complications. It is also important to note that not all institutions have 3D printing capabilities. In places where 3D printing is not readily available, Blum et al. proposed using several image processing tools in order to enhance the quality of images, such as 3D surface rendering, volume rendering, and global illumination rendering [34]. Simply saying, these techniques originate from the film and design industries whereby we are able to isolate areas or tissues of interests and apply mathematical formulas to determine the amount of lighting and exposure to various voxels of the CT images so as to present us with high-quality 3D models.

Nevertheless, our meta-analysis showed the possibility of combining technology with clinical applications to attempt to improve surgical and functional outcomes for patients. The reality of combining virtual reality with complex surgeries might not be a distant reality anymore. However, the evidence remains weak even with low-risk studies; therefore, it is not possible to conclude and identify if there are any clinically important differences between 3D printing and traditional approaches. Future studies should include a higher number of cases with randomization, adequate follow-up durations, and functional outcomes in order to further prove the efficacy of 3D printing and simulation. As for now, the decision to incorporate 3D printing and fracture mapping should be based on other considerations such as costs, levels of expertise, and surgeons’ preferences.

## Figures and Tables

**Figure 1 jcm-11-05258-f001:**
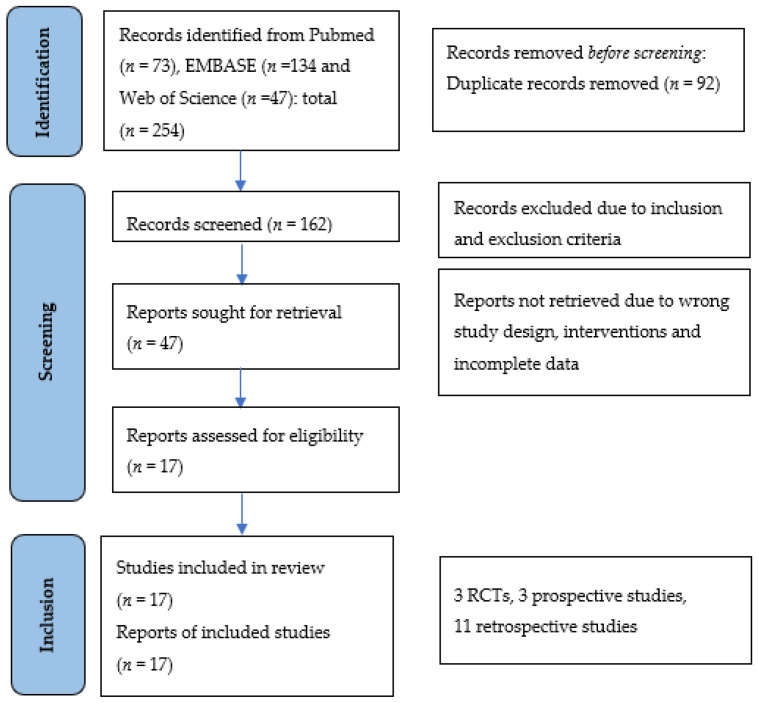
PRISMA flowchart for study selections. A total of 17 studies were included in our meta-analysis; of which 3 were randomized controlled trials (RCTs), 3 were non-randomized clinical trials, and 11 were retrospective studies. From: Page MJ, McKenzie JE, Bossuyt PM, Boutron I, Hoffmann TC, Mulrow CD, et al. The PRISMA 2020 statement: an updated guideline for reporting systematic reviews. BMJ 2021;372:n71. doi: 10.1136/bmj.n71.

**Figure 2 jcm-11-05258-f002:**
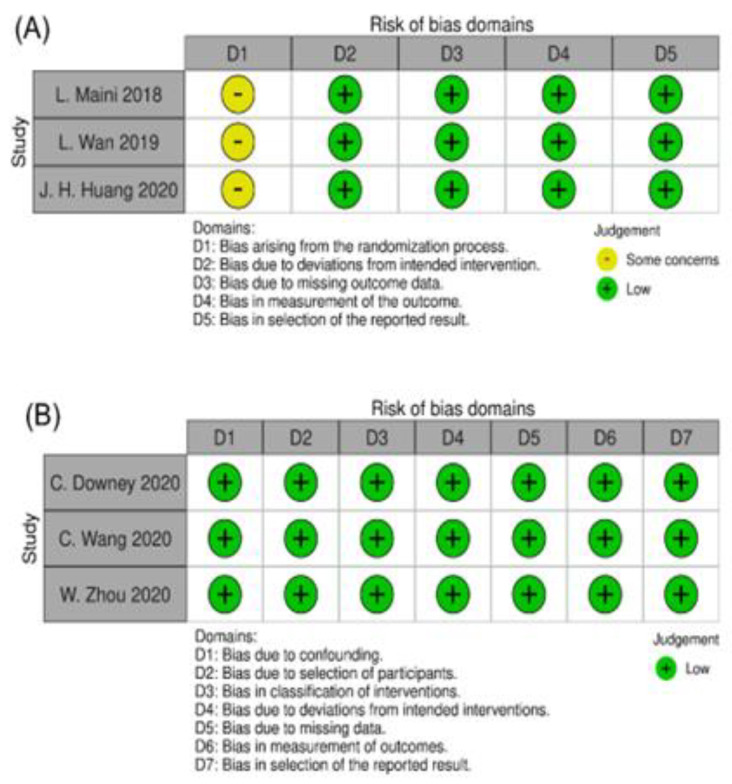
Risk of bias assessed with (**A**) revised Cochrane risk-of-bias tool for randomized controlled trials (RoB 2); (**B**) risk of bias in non-randomized studies-of interventions (ROBINS-I) assessment tool for non-randomized trials; and (**C**) risk of bias in non-randomized studies-of interventions (ROBINS-I) assessment tool for retrospective studies.

**Figure 3 jcm-11-05258-f003:**
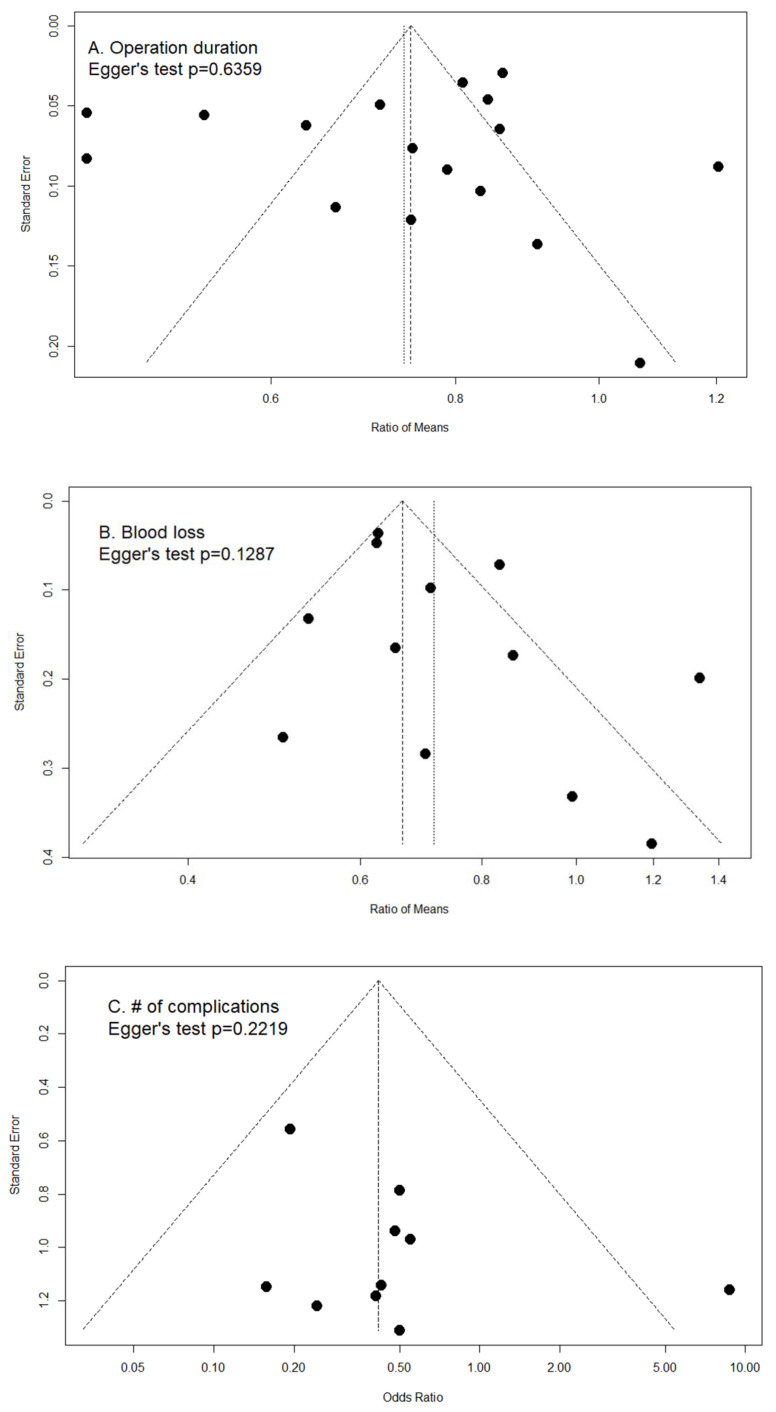
Funnel plots with pseudo 95% confidence intervals of (**A**) ratio of means in operation duration (Egger’s test *p* = 0.6359); (**B**) ratio of means in amount of blood loss (Egger’s test *p* = 0.1287); (**C**) odds ratio in number of complications (Egger’s test *p* = 0.2219); and (**D**) odds ratio in excellent/good reduction (Egger’s test *p* = 0.1230).

**Figure 4 jcm-11-05258-f004:**
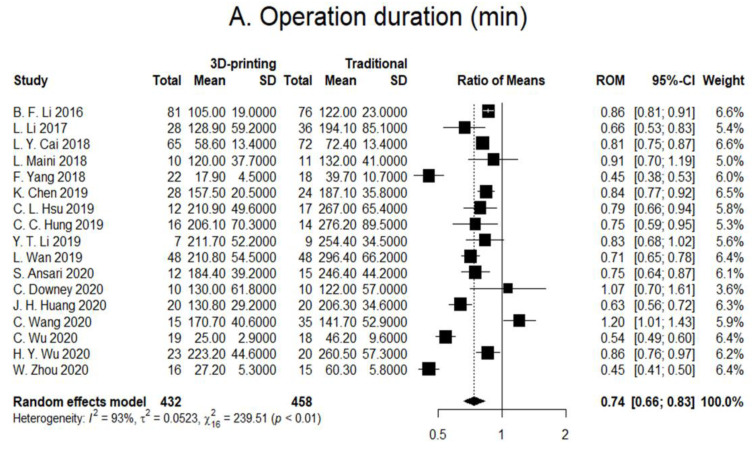
Forest plots of (**A**) surgical duration (minutes). Overall OR (95% CI) = 0.74 (0.66, 0.83); (**B**) amount of blood loss (mL). Overall OR (95% CI) = 0.71 (0.63, 0.81); (**C**) times of intraoperative imaging. Overall OR (95% CI) = 0.36 (0.17, 0.76); (**D**) number of complications. Overall OR (95% CI) = 0.42 (0.22, 0.78); (**E**) excellent/good reduction. Overall OR (95% CI) = 1.53 (1.08, 2.17).

**Figure 5 jcm-11-05258-f005:**
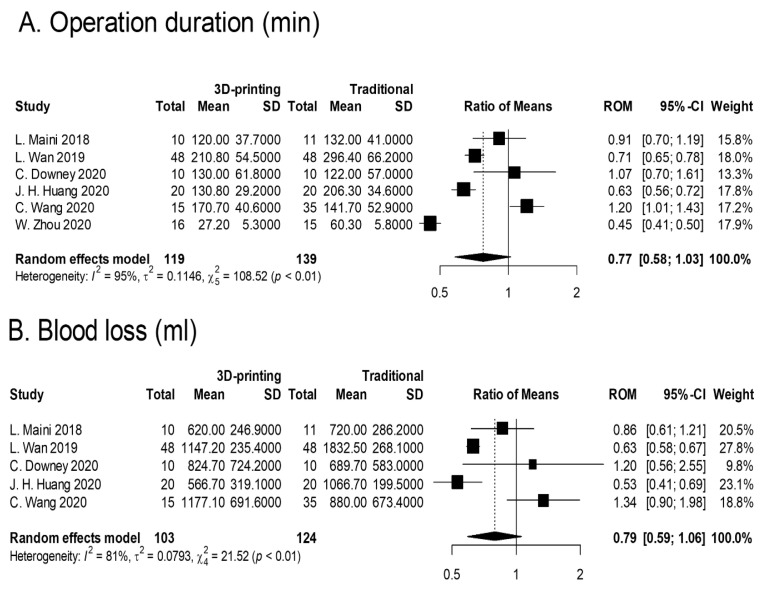
Forest plots of (**A**) surgical duration (minutes). Overall RoM (95% CI) = 0.77 (0.58, 1.03), I^2^ = 95%; (**B**) amount of blood loss (mL). Overall RoM (95% CI) = 0.79 (0.59, 1.06), I^2^ = 81%; (**C**) times of intraoperative imaging. Overall RoM (95% CI) = 0.32 (0.11, 0.93), I^2^ = 97%; (**D**) number of complications. Overall OR (95% CI) = 0.23 (0.10, 0.55), I^2^ = 0%; (**E**) excellent/good reduction. Overall OR (95% CI) = 2.00 (1.07, 3.73), I^2^ = 0%.

**Table 1 jcm-11-05258-t001:** Summary of findings (*n* = 17).

Study Characteristics	Design	Sample Sizes (Total/ Control/Treatment)	Population Characteristics	Fracture Mapping and 3D Printing Technology Used	Key Operative Outcomes	Duration of Follow-Ups
B.F. Li, 2016, China, [18]	Retrospective	157/76/81 Tile C pelvic fracture only Preoperative: CT	Age: Control: 32.2 ± 9.4 Treatment: 34.0 ± 8.2 Gender: Control: 51/25 Treatment: 55/26 ISS: Control: 19.6 ± 11.2 Treatment: 16.7 ± 14.7	3D image processing software: MIMICS (Materialise Interactive Medical Image Control System Software, Materialise, Belgium) 3D printed model: Fractured pelvis with soft tissues Preoperative: Pre-contouring of plates, approach, and entering position of screws.	Surgical related outcomes: Improved surgical duration (122 ± 23 vs. 105 ± 19), anesthesia duration (155 ± 26 vs. 135 ± 28), Hb (6.2 ± 7.4 vs. 31.0 ± 8.2), blood transfusion (9.9% vs. 19.7%), PONV (8% vs. 10%), postoperative pain (2.5 ± 2.3 vs. 2.8 ± 2.0), length of hospital stay (7.8 ± 2.0 vs. 10.2 ± 3.1). Functional outcome: more patients were discharged home rather than to rehabilitation	No long-term follow-ups
L. Li, 2017, China, [2]	Retrospective	64/36/28 Tile C pelvic fracture only Preoperative: X-ray and 2D CT	Age: Control: 34.5 ± 8.4 Treatment: 32.4 ± 7.6 Gender: Control: 28/8 Treatment: 18/10 ISS: Control: 17.2 ± 12.1 Treatment: 18.4 ± 15.3	3D image processing software: MIMICS (Materialise Interactive Medical Image Control System Software, Materialise, Belgium) 3D printed model: Fractured pelvis with no soft tissues. Preoperative: Pre-contouring of plates, selection of incision approach, and entering position of screws.	Surgical related outcomes: Improved surgical duration (128.9 ± 59.2 vs. 194.1 ± 85.1) blood loss (481.4 ± 103.2 vs. 771.29 ± 114.4), blood transfusion (345.1 ± 75.4 vs. 736.6 ± 125.9). Functional outcomes: Improved complications (1 vs. 3) Matta (excellent/good/poor: 20/6/2 vs. 8/16/12), Majeed (excellent/good/fair at 1 year: 20/6/2 vs. 18/7/11; 10 yr: 18/6/4 vs. 18/6/12)	Up to 10 years
F. Yang, 2018, China, [21]	Retrospective	40/18/22 Traumatic incomplete or complete disruptions of the posterior pelvic ring (types 54B and 54C) 54B/54C Control: 4/14 Treatment: 7/15 Preoperative: CT	Age: Control: 50.1 ± 13.7 Treatment: 51.7 ± 15.2 Gender: Control: 10/8 Treatment: 11/11 ISS: Control: 17.2 ± 12.1 Treatment: 18.4 ± 15.3	3D image processing software: MIMICS (Materialise Interactive Medical Image Control System Software, Materialise, Belgium) 3D printed model: Template guide Preoperative: Virtual simulation of screw placement with printing of screw template guide.	Surgical related outcomes: Improved surgical duration (17.9 ± 4.5 vs. 39.7 ± 10.7), amount of radioactive exposure (742.8 ± 230.6 vs. 1904.0 ± 844.5 cGy), rate of screw perforation (1 of 37 vs. 4 of 38). Functional outcomes: Reduction quality (excellent/good: 7/12 vs. 5/11).	No long-term follow-ups
** L. Maini, 2018, India, [14]	RCT	21/11/10 Control: 62A1 (1), 62A2 (1), 62A3 (1), 62B1 (2), 62B2 (5) 62C1 (1) Treatment: 62A1 (2), 62B1(4), 62B2 (3) 62C1 (1) Preoperative: CT	Mean age: 38.7 Gender: 18/21 ISS: Not mentioned	3D image processing software: MIMICS (Materialise Interactive Medical Image Control System Software, Materialise, Belgium) 3D printed model: Fractured pelvis with no soft tissues. Preoperative: Pre-contouring of plates.	Surgical related outcomes: Surgical duration (120 ± 37.7 vs. 132 ± 41) blood loss (620 ± 246.9 vs. 720 ± 286.2), additional instrumentation tines (9 vs. 0 min) Functional outcomes: Improved complications (1 vs. 2), CT residual displacement (4.75 ± 3.13 vs. 7.60 ± 4.92 mm), CT reduction (anatomic: 1 vs. 4).	No long-term follow-ups
L.Y. Cai, 2018, China, [22]	Retrospective	137/72/65 Tile B and C pelvic fracture Preoperative: CT	Age: Control: 32.63 ± 4.72 Treatment: 33.08 ± 4.91 Gender: Control: 45/27 Treatment: 37/28 ISS: Not mentioned	3D image processing software: MIMICS (Materialise Interactive Medical Image Control System Software, Materialise, Belgium) 3D printed model: Mirroring of contralateral healthy pelvis. Preoperative: Pre-contouring of plates and K-wires.	Surgical related outcomes: Surgical duration (58.63 ± 13.38 vs. 72.38 ± 13.4), timing of radioactive exposure (29.31 ± 2.83 vs. 36.63 ± 2.83). Functional outcome: Fracture healing time (13.8 ± 1.96 vs. 14.5 ± 1.56 weeks), Majeed (excellent: 80.6% vs. 78.5%) and Matta (excellent: 84% vs. 81.5%).	No long-term follow-ups
C.C. Hung, 2018, Taiwan, [15]	Retrospective	30/14/16 Control: Tile A (1), Tile B (8), Tile C (5) Treatment: Tile A (1), Tile B (6), Tile C (9) Preoperative: CT	Age: Control: 35.64 ± 17.37 Treatment: 35.44 ± 13.52 Gender: Control: 8/6 Treatment: 10/6 ISS: Not mentioned	3D image processing software: MIMICS (Materialise Interactive Medical Image Control System Software, Materialise, Belgium) 3D printed model: Mirroring of contralateral healthy pelvis Preoperative: Pre-contouring of plates and selection of screw length	Surgical related outcomes: Improved surgical duration (206.13 ± 70.32 vs. 276.21 ± 89.53), blood loss (45.6 ± 15.26 vs. 549.29 ± 404.43), instrumentation time (45.63 ± 15.26 vs. 102.86 ± 25.85). Functional outcome: Complications (1 vs. 3), CT reduction (good: 13 vs. 8).	2 years
C.L. Hsu, 2018, Taiwan, [11]	Retrospective	29/17/12 Control: Posterior wall (5), both columns (4), transverse with posterior wall (2), posterior column with posterior wall (1), T-shaped with posterior wall (4), both columns with posterior wall (1) Treatment: Posterior wall (3), posterior column (1), transverse (1), both columns (2), transverse with posterior wall (2), posterior column with posterior wall (1), T-shaped with posterior wall (1), both columns with posterior wall (1) Preoperative: CT	Age: Control: 38.24 ± 16.39 Treatment: 36.75 ± 16.39 Gender: Control: 14/3 Treatment: 11/1 ISS: Not mentioned	3D image processing software: MIMICS (Materialise Interactive Medical Image Control System Software, Materialise, Belgium) 3D printed model: Mirroring of contralateral healthy pelvis Preoperative: Surgical plan (including plate number, position, length, curvature, and screw position) and simulation of surgery with printed model.	Surgical related outcomes: Improved surgical duration (posterior column: 222.75 ± 48.12 vs. 259.76 ± 46.50), blood loss (anterior column: 433.33 ± 317.28 vs. 958.33 ± 427.10), instrumentation time (anterior column: 43.40 ± 10.92 vs. 99.00 ± 15.44; posterior column: 35.75 ± 9.21 vs. 67.35 ± 10.8). Surgical related outcomes: Surgical duration (anterior column: 199 ± 50.29 vs. 274.17 ± 80.95), blood loss (posterior column: 845.83 ± 681.06 vs. 866.47 ± 550.33) Functional outcome: Cmplications (2 vs. 5), X-ray reduction (good: 14 vs. 11).	Mean: 14.4 months (3–43 months)
K. Chen, 2019, China, [7]	Retrospective	52/24/28 T-shaped (10), anterior column with posterior hemi-transverse (16), double column (24). Preoperative: CT	Age: Control: 42.38 ± 12.28 Treatment: 46.11 ± 13.63 Gender: Control: 14/10 Treatment: 18/10 ISS: Not mentioned	3D image processing software: MIMICS (Materialise Interactive Medical Image Control System Software, Materialise, Belgium) 3D printed model: Mirroring of contralateral healthy pelvis. Preoperative: Pre-contouring of plates.	Surgical related outcomes: Improved surgical duration (157.5 ± 20.48 vs. 187.08 ± 35.81), blood loss (696.07 ± 166.54 vs. 833.75 ± 227.44). Functional outcome: Merle d’Aubigné (16.25 ± 1.65 vs. 15.83 ± 1.88) and Matta (excelent/good: 19/24 vs. 25/28).	Not mentioned
** L. Wan, 2019, China, [23]	Non-randomized controlled trials	96/48/48 Control: T-shaped (8), posterior column with posterior wall (11), both columns (9), transverse with posterior wall (12), anterior with transverse (6), marginal(1), compression fracture(1). Treatment: T-shaped (7), posterior column with posterior wall (12), both columns (8), transverse with posterior wall (11), anterior with transverse (7), marginal (2), compression fracture (2) Preoperative: CT	Age: Control: 41.88 ± 4.97 Treatment: 43.44 ± 4.53 Gwender: Control: 32/16 Treatment: 34/14 ISS: Not mentioned	3D image processing software: MIMICS (Materialise Interactive Medical Image Control System Software, Materialise, Belgium) 3D printed model: Both reduced and fractured pelvis models Preoperative: Simulation, selection of incision approach, sequence of reduction, placement of reduction clamp, rotation direction, placement of plate and screw, angle, length and pre-contouring of plates.	Surgical related outcomes: Improved surgical duration (150.24 ± 75.45 vs. 296. ± 76.83), blood loss (1147.2 ± 235.4 vs. 1832.5 ± 268), timing of radioactive exposure (6.8 ± 75.45 vs. 12.4 ± 2.1). Functional outcome: Improved complications (5 vs. 18). Functional outcomes: CT reduction (excellent: 81.25% vs. 77.08%), hip joint function (excellent: 87.5% vs. 83.3%)	Up to 6 months
Y.T. Li, 2019, Taiwan [19]	Retrospective	16/9/7 Control: T-shaped with posterior wall (2), posterior column with posterior wall (1), posterior wall (6) Treatment: T-shaped with posterior wall (1), posterior column with posterior wall (1), posterior wall (3), transverse with posterior wall (2) Preoperative: CT	Age: Control: 37.00 ± 17.09 Treatment: 32.14 ± 14.63 Gender: Control: 6/3 Treatment: 7/0 ISS: Not mentioned	3D image processing software: MIMICS (Materialise Interactive Medical Image Control System Software, Materialise, Belgium) 3D printed model: Mirroring of contralateral healthy pelvis onto fractured model simulation. Preoperative: Surgical plan (including the type of plate and plate number, curvature, position, and screw length) with pre-contouring of plates.	Surgical related outcomes: Surgical duration (211.71 ± 52.23 vs. 254.44 ± 34.46), blood loss (735.71 ± 614.22 vs. 742.22 ± 228.68). Surgical related outcomes: Improved instrumentation timing (38.43 ± 10.81 vs. 71.78 ± 9.69). Functional outcome: Complications (2 vs. 5), CT residual displacement (<2 mm: 7 vs. 7).	No long-term follow-ups
** C. Downey, 2020, Ireland, [10]	Non-randomized controlled trials	20/10/10 Anterior column posterior hemi-transverse (8), both columns (8), T-shaped (2), lateral compression complex (2). Preoperative: CT	Age: Control: 51.8 ± 14.9 Treatment: 51.9 ± 18.9 Gender: Control: 9/1 Treatment: 9/1 ISS: Control: 18.8 ± 3.8 Treatment: 20.6 ± 8	3D image processing software: Meshmixer 3D printed model: Fractured pelvis Preoperative: Pre-operative planning	Surgical related outcomes: Surgical duration (122 vs. 130), blood loss (689.7 vs. 824.7), radioactive exposure (727.1 vs. 1078.1 cGy), Functional outcome: similar EQ-5D-5L scores, reduction (anatomical: 10% vs. 10%) Others: Surgeon questionnaire	12 months
** C. Wang, 2020, China, [24]	Non-randomized controlled trials	50/35/15 Control: both columns (22), anterior column posterior hemi-transverse (10), T-shaped (3) Treatment: both columns (11), anterior column posterior hemi-transverse (3), T-shaped (1) Preoperative: CT	Age: Control: 45.1 ± 12.6 Treatment: 46.6 ± 12.3 Gender: Control: 22/13 Treatment: 10/5 ISS: Not mentioned	3D image processing software: MIMICS (Materialise Interactive Medical Image Control System Software, Materialise, Belgium) 3D printed model: Mirroring of the contralateral healthy acetabulum Preoperative: 3D printed plates and screws with simulation	Surgical related outcomes: Improved surgical duration (141.7 ± 52.9 vs. 170.7 ± 40.6), blood loss (880.0 ± 673.4 vs. 1177.1 ± 691.6). Functional outcome: Improved residual displacement (1.51 ± 0.97 vs. 2.38 ± 1.10 mm) Functional outcome: Complications (1 vs. 5), Matta (anatomical/satisfactory: 10/4 vs. 18/13).	Not mentioned
C. Wu, 2019, China, [25]	Retrospective	37/18/19 Control: Tile B (15), Tile C (3) Treatment: Tile B (15), Tile (4) Preoperative: CT	Age: Control: 42.4 ± 5.7 Treatment: 43.1 ± 12.7 Gender: Control: 12/6 Treatment: 12/7 ISS: Not mentioned	3D image processing software: MIMICS (Materialise Interactive Medical Image Control System Software, Materialise, Belgium) 3D printed model: Fractured pelvis model with template guide	Surgical related outcomes: Improved surgical duration per screw (25 ± 2.9 vs. 46.2 ± 9.6), radiation exposure (12.1 ± 4 vs. 56.1 ± 6.8). Functional outcome: Improved grading (I/II: 40%/2% vs. 26%/3%). Functional outcome: Reduction (excellent/good: 6/11 vs. 5/11).	6 months
H.Y. Wu, 2020, China, [20]	Retrospective	43/20/23 Double column acetabular fracture Preoperative: CT	Age: Control: 50.1 ± 8.2 Treatment: 51.0 ± 8.6 Gender: Control: 15/5 Treatment: 16/7 ISS: Not mentioned	3D image processing software: MIMICS (Materialise Interactive Medical Image Control System Software, Materialise, Belgium) 3D printed model: Mirroring of the contralateral healthy pelvis with template guide	Surgical related outcomes: Improved surgical duration per screw (260.5 ± 57.3 vs. 223.2 ± 44.6), blood loss (1426.1 ± 733.1 vs. 930.4 ± 523.2). Functional outcome: Complications (3 vs. 6), reduction (anatomical: 13 vs. 12), Merle d’Aubigné (excellent: 11 vs. 11), duration of hospital stay (24.6 ± 4.9 vs. 26.4 ± 7.2).	Not mentioned
** J.H. Huang, 2020, China, [5]	RCT	40/20/20 Double column acetabular fracture Preoperative: CT	Age: Control: 37.4 ± 12.7 Treatment: 43.4 ± 11.6 Gender: Control: 14/6 Treatment: 12/8 ISS: Not mentioned	3D image processing software: MIMICS (Materialise Interactive Medical Image Control System Software, Materialise, Belgium) 3D printed model: Reduced pelvis model Preoperative: Pre-contouring of plate and screw measurement.	Surgical related outcomes: Improved surgical duration (130.8 ± 29.2 vs. 206.3 ± 34.6), blood loss (500 vs. 1050), instrumentation time (32.1 ± 9.5 vs. 57.9 ± 15.1), radiation exposure (4.2 ± 1.8 vs. 7.7 ± 2.6 secs). Functional outcome: Complications (1 vs. 5), postoperative X-ray reduction (good: 13 vs. 8). Functional outcome: Improved bone union (14.48 ± 1.52 vs. 15.85 ± 1.56 weeks).	1 to 5 months
S. Ansari, 2020, India, [1]	Retrospective	27/15/12 Control: posterior column with posterior wall (6), transverse with posterior wall (4), double columns (3), anterior column posterior hemi-transverse (1), T-shaped (1) Treatment: posterior column with posterior wall (5), transverse with posterior wall (3), double columns (2), anterior column posterior hemi-transverse (1), T-shaped (1) Preoperative: CT	Age: Control: 39.1 ± 12.4 Treatment: 41.3 ± 13.7 Gender: Control: 12/3 Treatment: 11/1 ISS: Not mentioned	3D image processing software: Syngo.via VB40 software (Siemens, Munich, Germany) 3D printed model: Mirroring of the contralateral healthy pelvis Preoperative: Pre-contouring of plate and screw measurement and surgical approach.	Surgical related outcomes: Improved surgical duration (184.4 ± 39.2 vs. 246.4 ± 44.2), blood loss (664 ± 186.4 vs. 936 ± 198.4), radiation exposure (22 ± 5.6 vs. 62 ± 16.5 min). Functional outcome: Complications (2 vs. 4), reduction (<2 mm: 11 vs. 11), Harris (79.7 ± 13.7 vs. 83.4 ± 12.3).	Not mentioned
** W. Zhou, 2020, China, [3]	Non-randomized controlled trials	31/15/16 Control: Tile C1 (13), Tile C2 (2) Treatment: Tile C1 (13), Tile C2 (2) Preoperative: CT	Age: Control: 47.1 ± 0.5 Treatment: 47.2 ± 0.8 Gender: Control: 11/4 Treatment: 11/5 ISS: Not mentioned	3D image processing software: MIMICS (Materialise Interactive Medical Image Control System Software, Materialise, Belgium) 3D printed model: Reduced pelvis with template guide	Surgical related outcomes: Improved surgical duration per screw (27.2 ± 5.3 vs. 60.3 ± 5.8), radiation exposure (2.7 ± 0.5 vs. 15.4 ± 3.5). Functional outcome: Matta (excellent: 13 vs. 14), Majeed (excellent: 12 vs. 14).	6 to 20 months

RCTs and non-randomized controlled trials marked with (**). The surgical durations and radiation exposures were recorded as minutes and blood loss were recorded as millimeters unless as recorded. Preoperative imaging used in the respective studies were listed under the classifications. Legend: ISS = Injury severity score; RCT = randomized controlled trials; PONV = postoperative nausea/vomiting.

## Data Availability

Not applicable.

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
