# Peer review of "Three-Dimensional Printing and Fracture Mapping in Pelvic and Acetabular Fractures: A Systematic Review and Meta-Analysis"

_jcm, 2022, doi:10.3390/jcm11185258_

Round 1

Reviewer 1 Report

Introduction:  I know it is a small point, but instead of using the term “road accidents” I suggest using the word “road crashes”.  Many of the crashes in newer injury prevention thought is they are preventable (eg. use seat belts, not drive while impaired, improved road construction, etc.).

Similarly, use the word “sustain” instead of “suffers”. 

An updated meta-analysis “will be helpful” rather than “is required to validate”.

Last paragraph of Discussion, please delete the first sentence.  It is simply redundant.

Materials and Methods:

Study identification:  Please amplify what the wrong study designs and wrong interventions were.

Data synthesis and statistical analysis:  2nd to last sentence of 1st paragraph, what is meant by Matta, or postoperative x-ray.  I can’t quite follow.  Do you mean Matta scores?

Results:

For those papers with no mention of follow-up length, how can we be sure that the results regarding complications is appropriate/correct?  This is very concerning.  Should not those papers be deleted?  A complication such as loss of fixation, non-union could appear a few weeks or months after the surgery.  This needs to clearly explained.

T

Discussion:

2nd paragraph:  You state that fractures of the pelvis usually involve joint surfaces.  Can you please support this?  If you include all fractures of the pelvis, especially those in the elderly due to simple falls, I suspect that most will NOT involve joint surfaces.  Please support with a percentage number and reference.  Your reference 28 is only for acetabular fractures.

Author Response

Dear Reviewer,

We thank you for your precious time and effort in providing us with valuable feedbacks. 

Our replies to your suggestions and reviews are as provided in the Word Document attached.

Thank you.

Reviewer 2 Report

This work is a really intersting topic as 3D techniques are fastly and widely growing up. It brings several information concerning its benefits.

Here are a few comments that should allow to improve the manuscript

·        multiples headlines in table 1

·        material and methods:

·        did all the patients underwent radiographs and CT or only radiographs?

·        discussion paragraph 2: "Furthermore, the surgeons felt that the 3D models contributed to improved inter- and intra-specialty communication and enhanced overall team confidence during the surgery itself"

Did it significantly change treatment strategy? ie operative vs non operative and type of operative procedure

Did it change fracture classification? Is there any information available concerning inter observer reliability in acetabular fracture which are known to be variable using radiographs.

3D printing is not available in every institution. 3D imaging is available almost anywhere and could be considered as an alternative when printing ain't. Is there any information available concerning the help provided by 3D imaging? Because of the reasons above-mentioned, I think it should be discussed in the discussion section, as well as the imaging techniques used should be precised in the material and methods section.

Author Response

Dear Reviewer,

We thank you for your precious efforts in reviewing our article and providing us with very valuable feedbacks.

Our replies to your reviews and suggestions are as attached in the Word Document.

Thank you!

Round 2

Reviewer 1 Report

All of my queries have been answered.

Author Response

Thank you for taking precious time off your schedule to review our article.

Reviewer 2 Report

The discussion about 3D imaging without printing still has not been included in the paper. Volume rendering and global illumination are techniques that should be considered when 3D printing is not available and deserves to be mentionned as printing is not currently practiced, unlike CT post-treatment (cf 3D reconstructions, 4D imaging and postprocessing with CT in musculoskeletal disorders: Past, present and future - 07/10/20 Doi: 10.1016/j.diii.2020.09.008)

Author Response

We apologize for the oversight.

We had included your suggestion regarding CT processing and its reference into the second last paragraph of the discussion.

Thank you.
